# Detection and Classification of Knee Injuries from MR Images Using the MRNet Dataset with Progressively Operating Deep Learning Methods

Ali Can Kara *[ID] and Fırat Hardalaç

Department of Electrical and Electronics Engineering, Faculty of Engineering, Gazi University, Ankara TR 06570, Turkey; firat@gazi.edu.tr
* Correspondence: alicankara48@gmail.com; Tel.: +90-543-815-8346

**Abstract:** This study aimed to build progressively operating deep learning models that could detect meniscus injuries, anterior cruciate ligament (ACL) tears and knee abnormalities in magnetic resonance imaging (MRI). The Stanford Machine Learning Group MRNet dataset was employed in the study, which included MRI image indexes in the coronal, sagittal, and axial axes, each having 1130 trains and 120 validation items. The study is divided into three sections. In the first section, suitable images are selected to determine the disease in the image index based on the disturbance under examination. It is also used to identify images that have been misclassified or are noisy and/or damaged to the degree that they cannot be utilised for diagnosis in the first section. The study employed the 50-layer residual networks (ResNet50) model in this section. The second part of the study involves locating the region to be focused on based on the disturbance that is targeted to be diagnosed in the image under examination. A novel model was built by integrating the convolutional neural networks (CNN) and the denoising autoencoder models in the second section. The third section is dedicated to making a diagnosis of the disease. In this section, a novel ResNet50 model is trained to identify disease diagnoses or abnormalities, independent of the ResNet50 model used in the first section. The images that each model selects as output after training are referred to as progressively operating deep learning methods since they are supplied as an input to the following model.

**Keywords:** deep learning; transfer learning; ResNet50; convolutional neural networks; denoising autoencoders; knee injuries; magnetic resonance imaging; MRNet

## 1. Introduction

The application of artificial intelligence in the healthcare industry has grown substantially in recent years [1] since it may enhance diagnostic accuracy, boost efficiency in workflow and operations, and make monitoring of the patient's suffering easier. In the healthcare industry, computer-based technology is provided through collecting digitised data in fields [2], such as computed tomography (CT) [3], magnetic resonance imaging (MRI), X-ray [4], and ultrasound [5,6]. However, most of the best examples of the high performance of deep learning are in the field of computer vision by examining medical images and videos [7]. With breakthroughs in deep learning and image processing [8,9], it has the potential to recognise and locate complex patterns from several radiological imaging modalities, many of which even have recently demonstrated performance comparable to human decision making [1]. When reading medical images, radiologists frequently scan the entire image to locate lesions, analyse and quantify their attributes, and then define them in the report. This normal procedure takes a long time. More critically, certain important abnormal outcomes may go unnoticed by human readers [1]. Technological advancements have made it possible to generate high-resolution magnetic resonance (MR) images in a short amount of time, allowing for faster scanning. The employment of deep

learning approaches to diagnose utilising MRI data from diverse parts of the body is very common [10–13].

MRI is a technique for mapping the interior structure of the body as well as specific aspects of functioning. The MR image is produced by keeping the patient inside a gigantic magnet that induces a relatively powerful external magnetic field [14]. Geometric planes used to section the body into pieces are called body planes. They are commonly used to define the placement or orientation of bodily structures in both human and animal anatomy. Anatomical terminology refers to reference planes as standard planes. The body is divided into dorsal and ventral portions by the coronal plane (front or Y-X plane). The axial plane (axial or X-Z plane) divides the body into superior and inferior (head and tail) portions. It is usually a horizontal plane that is parallel to the ground and runs through the centre of the body. The sagittal plane (lateral or Y-Z plane) divides the body into sinister and dexter sides [15]. Figure 1 shows these axes.

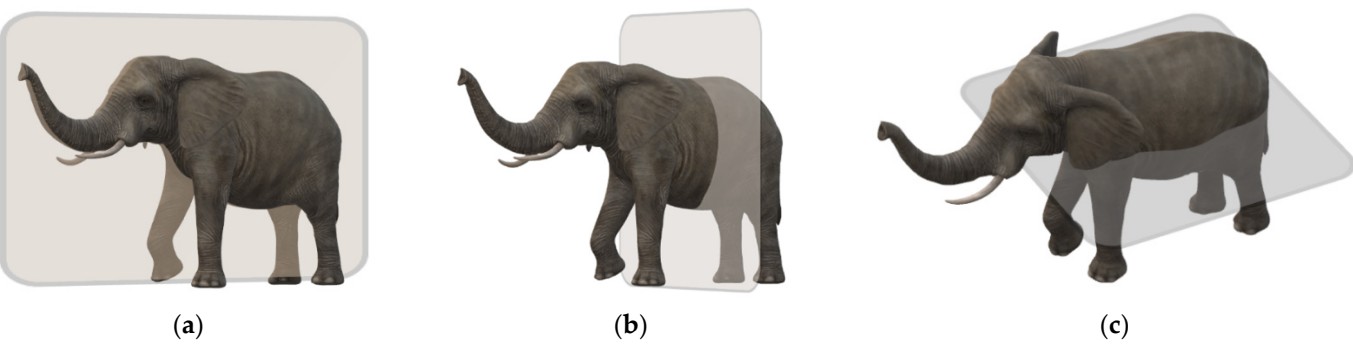

**Figure 1.** (**a**) Sagittal plane; (**b**) coronal plane; (**c**) axial plane [16].

We introduce a real-world MRI-based knee diagnostic system in this study. This study contributes to the detection and separation of misclassified or over noisy magnetic resonance (MR) images utilised in the hospital setting, as well as the designation of the relevant diagnostic region and, eventually, the diagnosis of the disease. Section 2 focuses on case studies involving the employment of deep learning and artificial intelligence in the healthcare industry. Additionally, further research on the issue and the methods that are employed in the studies are discussed as a consequence of the literature search. Furthermore, the methods we employed, as well as the architecture of the models we adopted in our study, are detailed in the following sub-sections of Section 2. Our findings are presented in Section 3. Section 4 provides the outcomes of other studies on the issue, and comparisons are made with the data we acquired. Section 5 outlines the benefits and drawbacks of the study we accomplished so far, as well as providing examples of possible future studies.

## 2. Materials and Methods

### 2.1. Dataset

The MRNet dataset [17] contains 1370 MRI examinations of knees accomplished at Stanford University Medical Centre. Each case is labelled based on whether the implicated knee had an anterior cruciate ligament (ACL) tear, a meniscal tear, or other indications of abnormalities. Different ailments may coexist in each examination. A patient suffering from an ACL tear, for example, may also have a meniscus condition. Each condition is represented by axial, coronal, and sagittal MR images in the analysed dataset. Each image is 256 × 256, and the number of slices ranges from 17 to 61. There are 1104 abnormal, 319 ACLs, and 508 menisci in the dataset [18]. The dataset contains 1130 trains, 120 valid, and 120 test datasets. We utilised 1130 trains and 120 validation data in our study.

### 2.2. Related Works

A considerable quantity of research has been published on artificial intelligence and healthcare due to the complexity of the work that a person has to do, the availability of labelled data, and the performance of deep learning in computer vision studies in order to detect disturbance in various subjects in the healthcare industry. The number of skin cancer cases, as an example to these research works, has been steadily growing over the last 30 years [19,20]. Studies were conducted by Ameri [21], employing AlexNet, one of the deep learning methods, and by Moldovanu et al. [22], using the feedforward backpropagation network (FFBPN) to locate benign and malignant skin lesions. Moldovanu et al. [23] also proved that integration of the fractal analysis approach to the standard colour distribution analysis for the location and classification of skin lesions boosts accuracy. Machine and deep learning approaches can be employed to automate location and classify a brain tumour. Many researchers have studied machine and deep learning algorithms for classifying different types of brain tumours [24,25]. Kempen et al. [26], for example, presented a comprehensive study analysis of the most prevalent glioma tumour in the brain, which employed machine learning approaches. In addition, Nasor and Obaid [27] developed the diagnosis and localisation of brain tumours through the combination of k-means clustering, object counting and patch-based image processing assessment techniques with computer vision methods. Breast cancer is also one of the most prevalent cancers among women. The use of MRI for early detection of breast cancer significantly contributes to patients' recovery. Artificial intelligence has excelled in image recognition tasks and is extensively researched in the field of breast cancer screening [28]. Benign and malignant tumours on MRI image sequences were classified by Hu et al. [29] through training the VGG19 model and support vector machine pre-trained with ImageNet, as well as by Ellman et al. [30], using the polynomial kernel function support vector machine model.

In the study by Bien et al. [17], each two-dimensional MR image slice was passed through a feature extractor based on AlexNet [31] to obtain an $s \times 256 \times 7 \times 7$ tensor, containing features for each slice. The loss of a sample was scaled inversely proportionate to the prevalence of that sample's class in the dataset to factor in the imbalanced class sizes across all tasks. Each training sample was randomly rotated between $-25$ and 25 degrees, randomly shifted between $-25$ and 25 pixels, and flipped horizontally with 50% probability when it appeared in the training.

In order to train the networks, Azcona et al. [18] used the ResNet18, ResNet50, and ResNet152 [32] models to compute the probability of ACL tear, meniscus tear, and knee abnormalities. They devised procedures to ensure that all patients' slices were entered into the network in the same number. Two procedures were tested for this, the first of which was to utilise only the centre slices of all the data from all the patients. This approach intends to assume that the slices in the centre of the sequence will likely contain more substantive data than those at the beginning and end. The total number of images obtained was selected as 17 since all patients had a minimum of 17 images. The second procedure involves running simple mathematical operations on a constant number of images to interpolate up or down. The outcomes of the second procedure appeared to be superior to those of using centre slices. A total of 15 interpolated slices were discovered to be the optimal constant number to interpolate.

Tsai et al. [33] applied an optimised efficiently layered network (ELNet) architecture by employing MRI to diagnose knee disturbances. The most notable contribution of this study is the network that introduces a novel slice feature that incorporates multi-slice normalisation with BlurPool down sampling. Three imaging were oriented for each examination in the MRNet dataset. The coronal images were employed in the study to detect meniscal tears, whereas axial images were utilised to detect ACL tears and abnormalities.

Roblot et al. [34] studied with a training dataset collected from 41 hospitals in France, including 1123 MR images in the training dataset and 700 images in the test dataset of the knee. The goal was to design an algorithm that can detect the presence of a tear in the meniscus, as well as the position and orientation of the tear. In each of the three stages,

CNN models were utilised. These methods are region-based convolutional neural network (R-CNN) and fast R-CNN [35] models.

Couteaux et al. [36] aimed to locate the meniscus and identify the tears in the meniscus. In both stages of the study, mask R-CNN [37] was employed.

Utilising deep learning approaches, Awan et al. [38] aimed to detect anterior cruciate ligament injury at an early stage through efficient and comprehensive automated MR imaging without engaging radiologists. This paper proposed a customised ResNet14 convolutional neural network architecture with six alternative orientations, as well as class-balancing and data augmentation. A total of 917 knee sagittal planes from a clinical medical centre were examined as a dataset.

### 2.3. Selecting Eligible Data

The adoption of standardised datasets and benchmarks for assessing machine learning algorithms has been a crucial factor in the advancement of computer vision. The scope of the tasks being addressed is defined by these datasets, which frequently include imaging data with human comments [7]. Transfer learning aims to maximise learning performance in the targeted field by storing acquired knowledge while addressing a problem and then transferring that knowledge that has been found in different but related fields. This may make it possible to avert the necessity for a large amount of data for the field targeted to be trained [39]. ImageNet [40] is a high-profile example of a large imaging dataset utilised to benchmark and examine the relative performance of machine learning models in the still image dataset field [41].

When examining the studies on modern convolutional neural networks, LeNet [42], developed by LeCun et al. in 1998; AlexNet [31], created by Krizkevsky et al. in 2012; VGGNet [43], produced by Simonyan and Zisserman at the Visual Geometry Group (VGG) laboratory in 2014; and residual networks (ResNet), built by He et al. in 2015, can be cited. Prior to ResNet, it was believed that increasing the number of convolutional layers would enhance the performance rate. On the contrary, increasing the network depth led to the vanishing gradient problem. The vanishing gradient issue appears when the impact of the initial layers is considerably weakened during backpropagation since the impact of the multipliers is drastically diminished. This problem is addressed by establishing shortcut connections between layers in the ResNet model. The vanishing gradient problem is handled by using these shortcuts to transfer the gradient values in the model after the residual blocks.

Multiple images are added one after the other to make up the MRI data. Selecting the eligible images to diagnose the disease while working on these images will improve the efficiency of the study. In addition, the accuracy of the classification process was checked, and the diagnosis of over noisy and damaged images were identified while selecting the eligible images. When the designed models were applied to the dataset during the classification process, it was discovered that some patients' data were not selected at all. When reviewed, the classification procedures appeared to be followed incorrectly, and the images were noisy or damaged. The following sections explain why some images in the dataset were not selected, as well as providing examples.

The ResNet50 model, which we studied in this section and was pre-trained through ImageNet, is suitable for working with three-channel images [44]. The images in the MRNet dataset have the sizes $256 \times 256 \times 1$. Therefore, one-channel images must be converted to three-channel ones. During this conversion, we built a function to convert all the images we had into three-channel images. Every time a new one-channel visual arrives at a function, the colour value of each pixel is repeated three times in an empty matrix of $256 \times 256 \times 3$ dimensions within the function to generate a new image, according to the working logic of the function. Figure 2 demonstrates the conversion of a one-channel image to a three-channel, as well as the function we employ for this conversion.

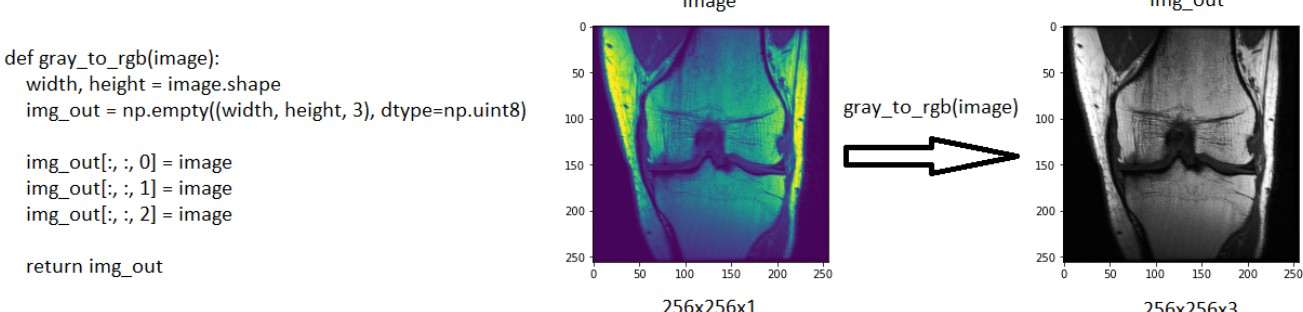

```
def gray_to_rgb(image):
    width, height = image.shape
    img_out = np.empty((width, height, 3), dtype=np.uint8)

    img_out[:, :, 0] = image
    img_out[:, :, 1] = image
    img_out[:, :, 2] = image

    return img_out
```

**Figure 2.** The function used to increase the number of channels and, an example of a $256 \times 256 \times 1$ coronal image, which was converted to $256 \times 256 \times 3$.

This section studied the ResNet50 model, which was pre-trained with ImageNet. In the model, the global average pooling and layer trainable were both set to true. The exponential decay learning rate was employed to develop a learning rate that drops as the training progresses. The initial learning rate was initially set at 0.01 and declined with a decay rate of 0.96 at 4000 decay steps. Stochastic gradient descent (SGD) was used as the optimiser. It is known that SGD has large oscillations [45]. Li et al. [46], on the other hand, demonstrated the SGD oscillation variation when the exponential decay is employed and utilised with different momentum values. The momentum value in our study was set to 0. However, the Keras resources [47], an artificial intelligence library that we utilised in our study, indicated the usage of exponential decay with SGD as an optimiser [48] and reported the momentum value of SGD was automatically assigned to be 0 by default [49]. Sagittal data were split into four different groups, coronal data into seven, and axial data into five. Figure 3 shows the model's structure as well as its parameter values.

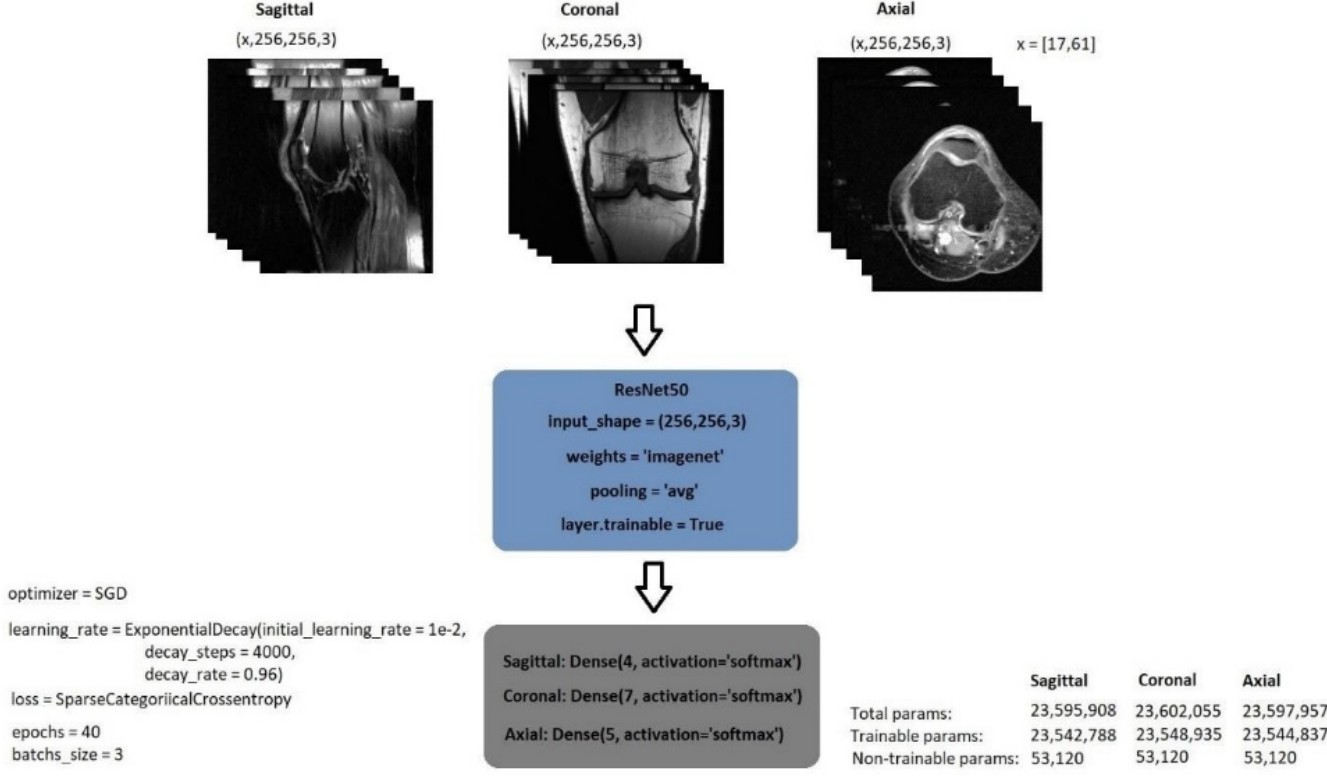

**Figure 3.** General structure of image classification model.

### 2.3.1. Classification of Sagittal Images and Results

Sagittal images were separated into four different classes during the classification process, with 50 patients from the train dataset being utilised, 40 of which were for training and 10 for testing. Sagittal meniscus data were selected by Class 2 among the formed classes, whereas Class 3 was used to select sagittal ACL data. Due to the disproportionate quantity of data in the classes, the weight values in the classification process were manually set during the training phase by utilising the sample weight values. Table 1 shows the train and test data quantities as well as sample weights for the four different classes that were formed.

**Table 1.** The sample weights values that were assigned inversely proportional to the number of elements in each class in the train and validation datasets created from sagittal image indexes and the number of elements in the train set are named Class 2, which is assigned to determine the sagittal ACL classes among these classes, and Class 3, which is assigned to determine the sagittal meniscus images.

| Sagittal Plain | Class 0 | Class 1 | Class 2 (ACL) | Class 3 (Meniscus) |
|---|---|---|---|---|
| Train | 219 | 180 | 515 | 236 |
| Validation | 45 | 43 | 117 | 55 |
| Sample Weights | 2.35 | 2.86 | 1.0 | 2.18 |

Figure 4a depicts the classification of images pertaining to a patient in a classification process; Figure 4b shows the confusion matrix produced after the training of sagittal images as a result of the data classification test of 10 patients, while Figure 4c provides the accuracy of 40 epochs and Figure 4d gives the train and validation loss values.

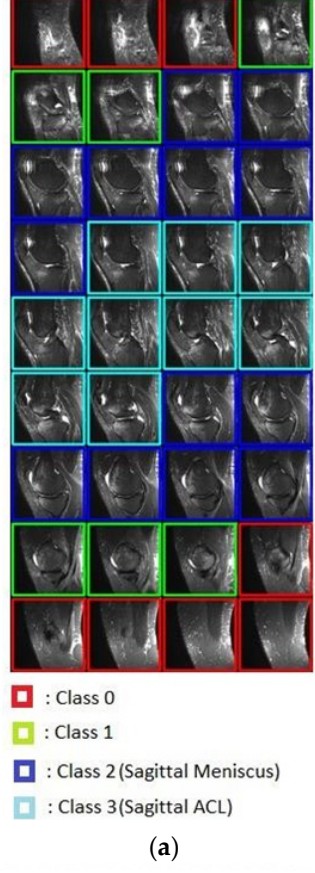

□ : Class 0
□ : Class 1
□ : Class 2 (Sagittal Meniscus)
□ : Class 3 (Sagittal ACL)

(**a**)

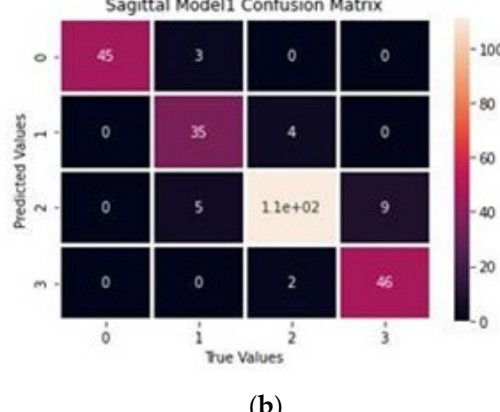

(**b**)

**Figure 4.** *Cont.*

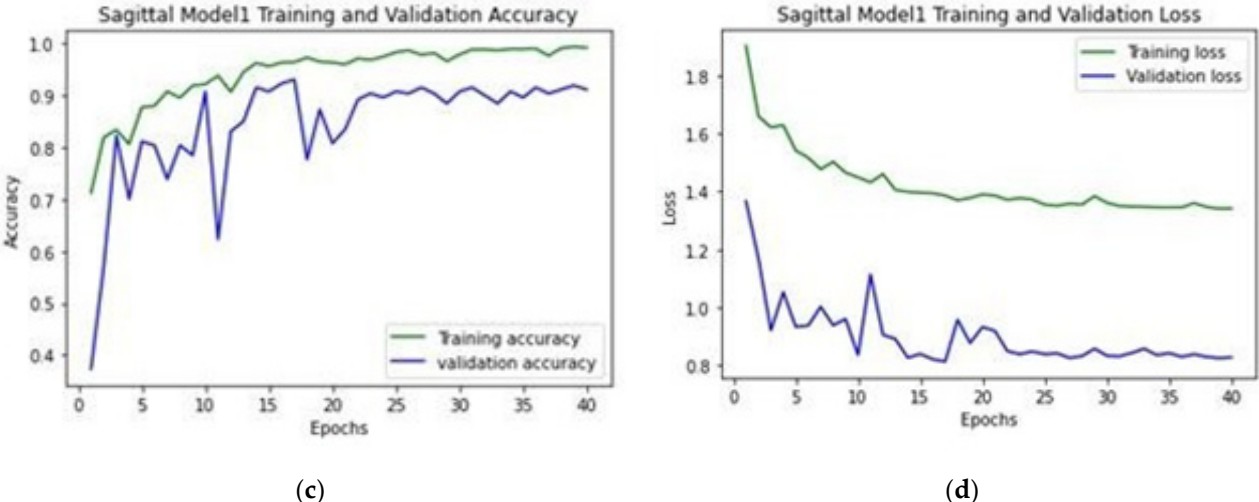

(**c**) (**d**)

**Figure 4.** (**a**) Classification process of sagittal MRI; (**b**) confusion matrix of the produced validation dataset for sagittal plane; (**c**) training and validation accuracy rates; (**d**) training and validation loss values.

Upon the examination of sagittal train and valid images, six trains and two valid data appeared to be not selected. The unselected images were observed to be misclassified and over noisy images. Table 2 contains a list of unselected images, and Figure 5 shows one example of unselected images.

**Table 2.** The number of images and a list of unselected images.

| Sagittal Plane | Total | Utilised | Unselected | Unselected List |
|---|---|---|---|---|
| Train | 1130 | 1124 | 6 | 0003, 0370, 0544, 0582, 0665, 0776 |
| Validation | 120 | 118 | 2 | 1159, 1230 |

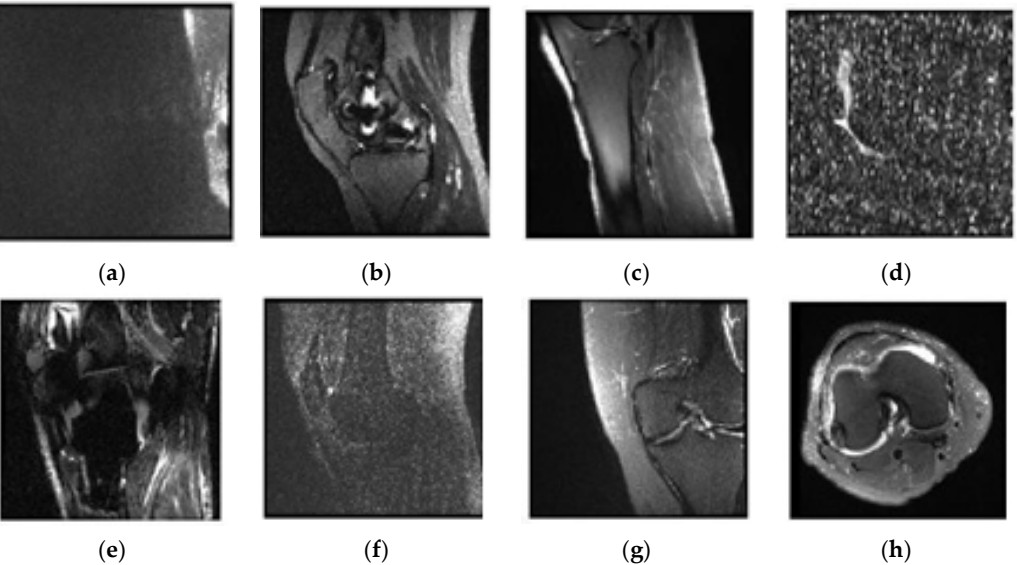

(**a**) (**b**) (**c**) (**d**)

(**e**) (**f**) (**g**) (**h**)

**Figure 5.** Examples of unselected images and patients' numbers in the dataset: (**a**) 0003; (**b**) 0370; (**c**) 0544; (**d**) 0582; (**e**) 0665; (**f**) 0776; (**g**) 1159; (**h**) 1230.

### 2.3.2. Classification of Coronal Images and Results

Coronal images were separated into seven different classes during the classification process, with 50 patients from the train dataset being utilised, 40 of which were for training

and 10 for testing. Among the formed classes, Class 3 was used to select coronal ACL and meniscus data. Due to the disproportionate quantity of data in the classes, the weight values in the classification process were manually set during the training phase by utilising the sample weight values. Table 3 shows the train and test data quantities as well as sample weights for the seven different classes that were formed.

**Table 3.** The sample weights values that have been assigned inversely proportional to the number of elements in each class in the train and validation datasets created from coronal image indexes and the number of elements in the train set are named Class 3, which is assigned to determine the coronal ACL and meniscus images.

| Coronal Plane | Class 0 | Class 1 | Class 2 | Class 3 (ACL and Meniscus) | Class 4 | Class 5 | Class 6 |
|---|---|---|---|---|---|---|---|
| Train | 139 | 139 | 144 | 188 | 126 | 139 | 354 |
| Validation | 44 | 22 | 36 | 56 | 44 | 28 | 89 |
| Sample Weights | 2.55 | 2.55 | 2.46 | 1.88 | 2.81 | 2.55 | 1.0 |

Figure 6a depicts the classification of images pertaining to a patient in a classification process; Figure 6b shows the confusion matrix produced after the training of coronal images as a result of the data classification test of 10 patients, while Figure 6c provides the accuracy of 40 epochs and Figure 6d gives the train and validation loss values.

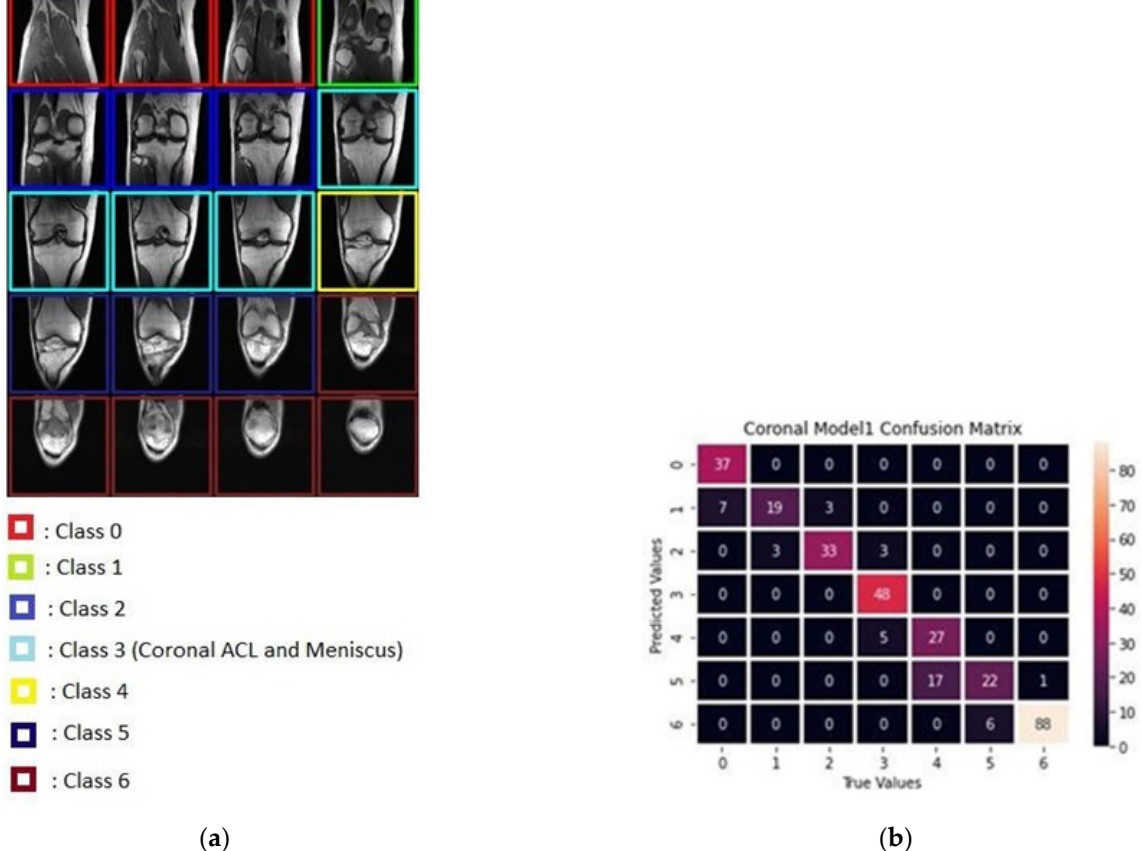

(**a**)                                                                                                  (**b**)

**Figure 6.** *Cont.*

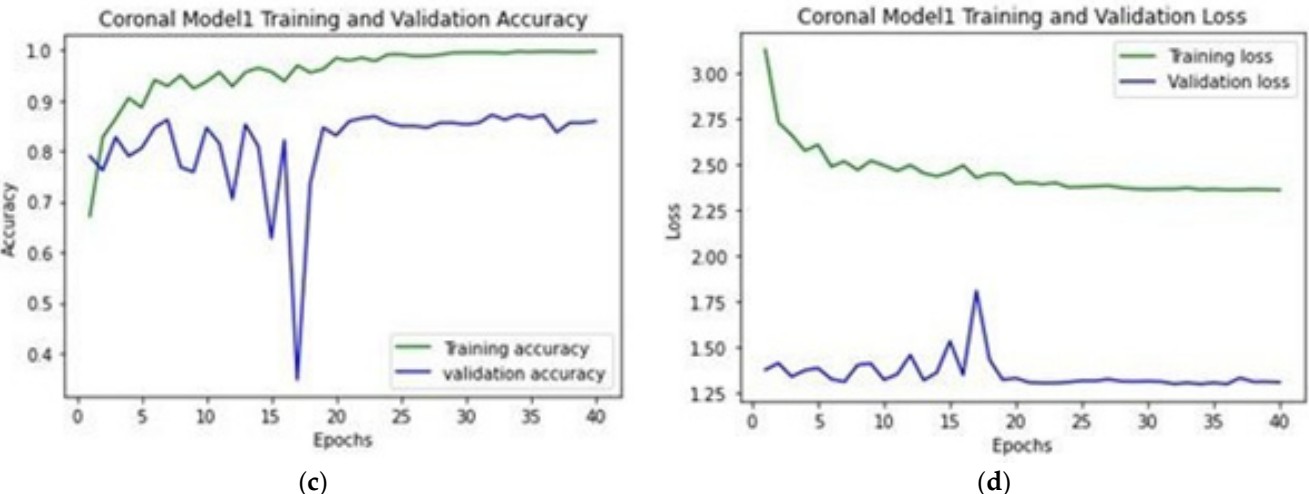

(**c**)                     (**d**)

**Figure 6.** (**a**) Classification process of coronal MRI; (**b**) confusion matrix of the produced validation dataset for coronal plane; (**c**) training and validation accuracy rates; (**d**) training and validation loss values.

Upon examination of the coronal train and validation images, seven train data appeared to be not selected. The unselected images were observed to be misclassified images. Table 4 contains a list of unselected images, and Figure 7 shows one example of unselected images.

**Table 4.** The number of images and a list of unselected images.

| Coronal Plane | Total | Utilised | Unselected | Unselected List |
|---|---|---|---|---|
| Train | 1130 | 1123 | 7 | 0310, 0544, 0610, 0665, 0975, 1010, 1043 |
| Validation | 120 | 120 | 0 | - |

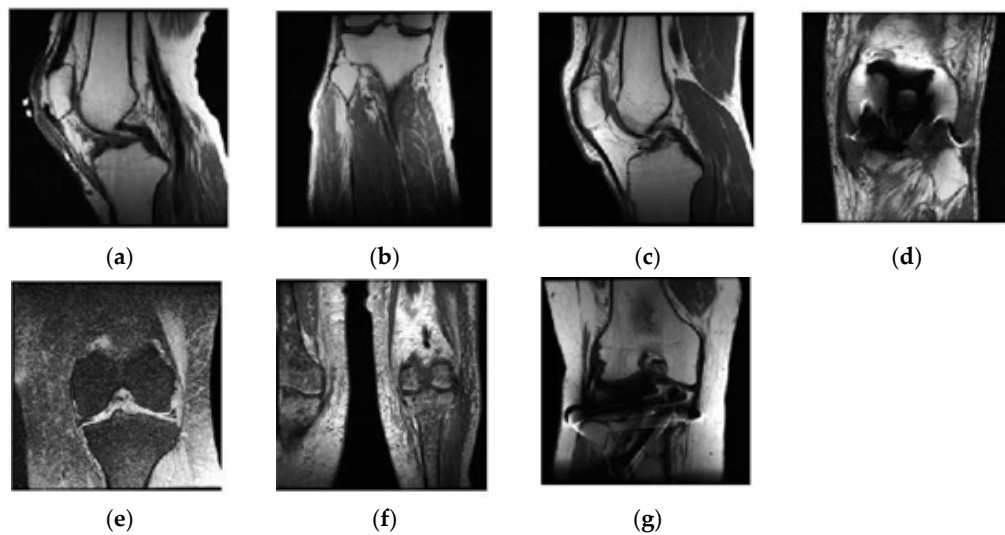

(**a**)        (**b**)        (**c**)        (**d**)

(**e**)        (**f**)        (**g**)

**Figure 7.** Examples of unselected images and patients' numbers in the dataset; (**a**) 0310; (**b**) 0544; (**c**) 0610; (**d**) 0665; (**e**) 0975; (**f**) 1010; (**g**) 1043.

2.3.3. Classification of Axial Images and Results

Axial images were separated into five different classes during the classification process, with 50 patients from the train dataset being utilised, 40 of which were for training and 10 for testing. Axial ACL data were selected by Class 1 among the formed classes, whereas Class 2 was used to select axial meniscus data. Due to the disproportionate quantity of

data in the classes, the weight values in the classification process were manually set during the training phase by utilising the sample weight values. Table 5 shows the train and test data quantities as well as sample weights for the five different classes that were formed.

**Table 5.** The sample weight values that were assigned inversely proportional to the number of elements in each class in the train and validation datasets created from axial image indexes and the number of elements in the train set are named Class 1, which is assigned to determine the axial ACL classes among these classes, and Class 2, which is assigned to determine the axial meniscus images.

| Axial Plane | Class 0 | Class 1 (ACL) | Class 2 (Meniscus) | Class 3 | Class 4 |
|---|---|---|---|---|---|
| Train | 508 | 327 | 162 | 166 | 281 |
| Validation | 122 | 81 | 37 | 35 | 51 |
| Sample Weights | 1.0 | 1.55 | 3.14 | 3.06 | 1.81 |

Figure 8a depicts the classification of images pertaining to a patient in a classification process; Figure 8b shows the confusion matrix produced after the training of axial images as a result of the data classification test of 10 patients, while Figure 8c provides the accuracy of 40 epochs, and Figure 8d gives the train and validation loss values.

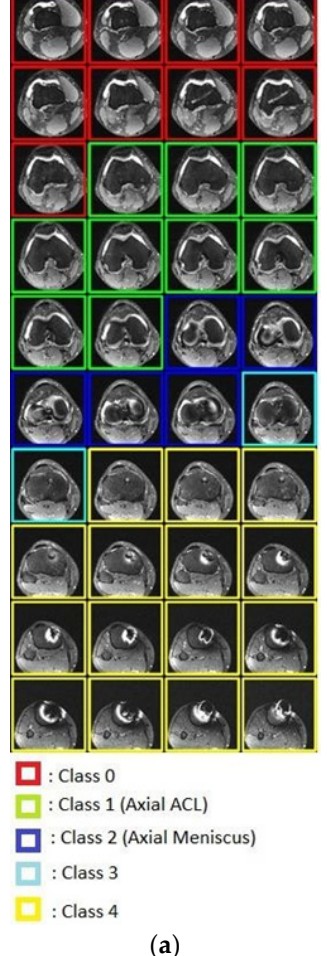

: Class 0
: Class 1 (Axial ACL)
: Class 2 (Axial Meniscus)
: Class 3
: Class 4

(**a**)

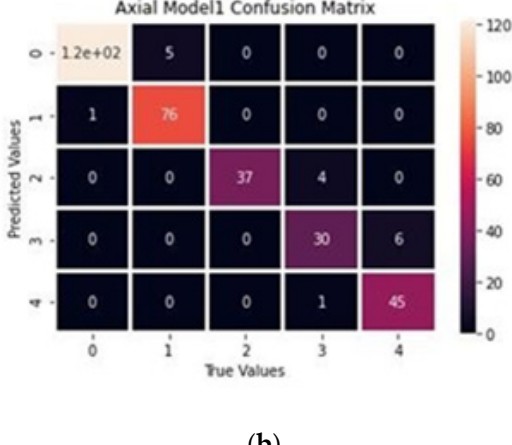

(**b**)

**Figure 8.** *Cont.*

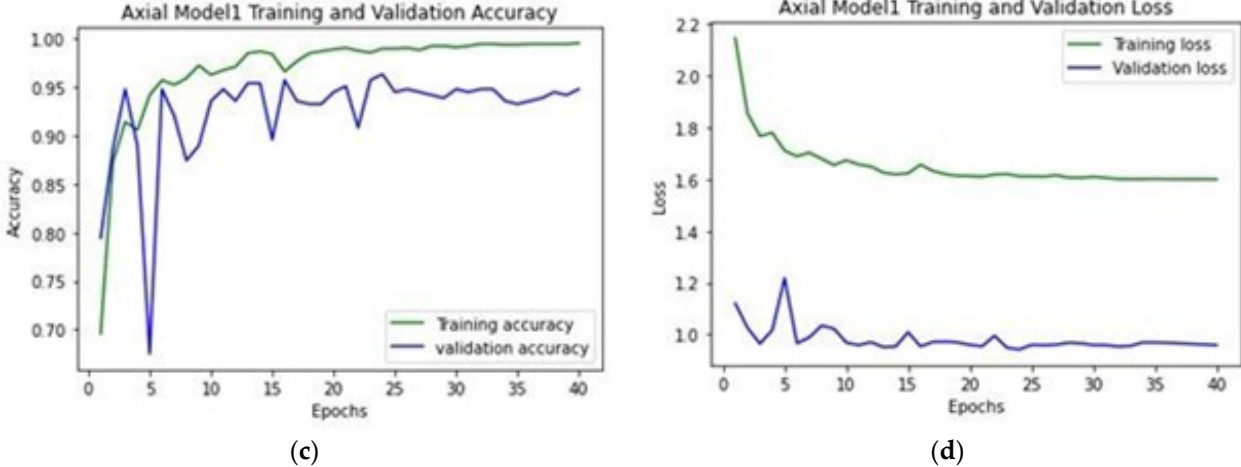

(**c**)          (**d**)

**Figure 8.** (**a**) Classification process of axial MRI; (**b**) confusion matrix of the produced validation dataset for axial plane; (**c**) training and validation accuracy rates; (**d**) training and validation loss values.

Upon examination of axial train and valid images, two trains and one valid data appeared to be not selected. The unselected images were observed to be damaged, over noisy, or ineligible for control. Table 6 contains a list of unselected images, and Figure 9 shows one example of unselected images.

**Table 6.** The number of images and a list of unselected images.

| Axial Plane | Total | Utilised | Unselected | Unselected List |
|---|---|---|---|---|
| Train | 1130 | 1128 | 2 | 0577, 0665 |
| Validation | 120 | 119 | 1 | 1136 |

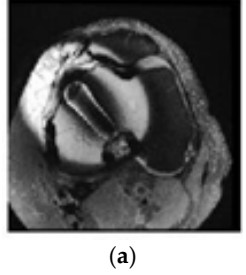 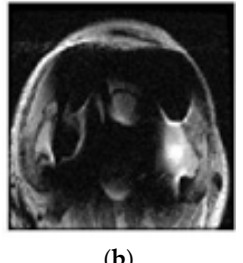 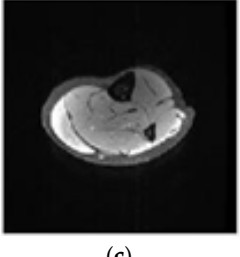

(**a**)            (**b**)            (**c**)

**Figure 9.** Examples of unselected images and patients' numbers in the dataset: (**a**) 0577; (**b**) 0665; (**c**) 1136.

*2.4. Selecting the Relevant Area*

Selecting the relevant regions would improve the accuracy of the study by focusing on the region where the disease will be diagnosed rather than looking at the entire image during the diagnostic phase. The study by Saygılı et al. [50] aimed to search the meniscus structures in a completely narrowed area with regard to this, which they performed by manually marking the MR images. They analysed a dataset that included both healthy and unhealthy MR images with varying degrees of discomfort. The dataset they employed contained 88 MR images that were labelled by radiologists. The study investigated the effect of feature extraction methods of histograms of oriented gradients (HOG) [51] and local binary pattern (LBP) [52]. HOG and LBP are feature extraction methodologies that produce successful results. HOG is a feature extraction methodology that delivers accomplished image recognition results. In this method, the feature is extracted with the gradients and orientations of the pixels on the image. Our study proposes a novel model for region

selection. By integrating our CNN and denoising autoencoder models, we propose a novel technique. This study aimed to correlate the features that were selected among the image by CNN with the marked regions in the output by removing the most distinctive parts of the denoising autoencoder model. The following sections describe how images are selected from the sagittal, coronal, and axial train datasets, as well as how data are augmented to train the model we built.

### 2.4.1. Selecting Relevant Areas on the Sagittal Axis

Among the 30-patient data in the sagittal train dataset, 30 images for the ACL and 30 images for the meniscus were selected during the selection of relevant regions, and these two circumstances went through a training procedure separately. The images selected for the ACL procedure were set as (60,100) dimensions, and the images selected for the meniscus were determined in (50,100) dimensions. Figure 10 indicates the examples from the manually selected regions. The original and marked images in the ACL data shifted four times by 10 pixels to the left and right; then, a new list of 9 images for each image and 270 images for a total of 30 images produced was scrolled up twice by 25 pixels and down once by 25 pixels, and a dataset of 1080 images was obtained. The original and marked images in the Meniscus data shifted four times by 10 pixels to the left and right; then, a new list of nine images for each image and 270 images for a total of 30 images produced was scrolled up twice by 25 pixels and down once by 25 pixels, and a dataset of 1080 images was obtained.

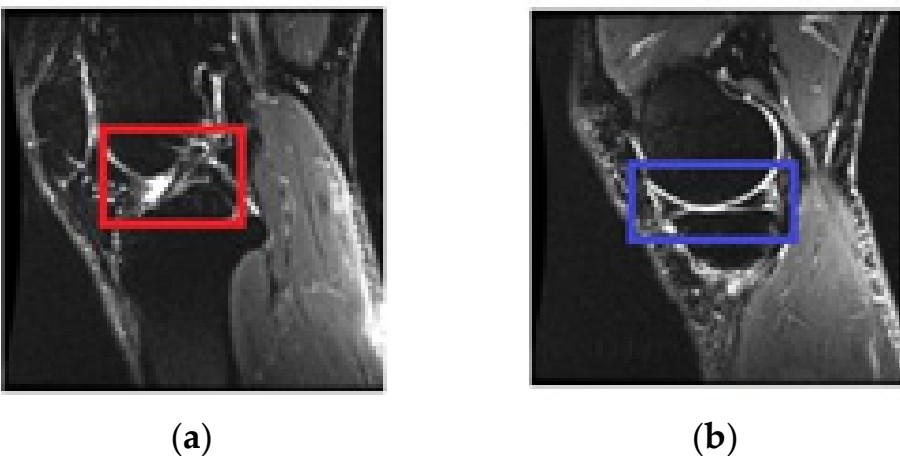

(**a**)                                          (**b**)

**Figure 10.** (**a**) The red square indicates the area selected for ACL diagnosis; (**b**) the blue square indicates the area selected for meniscus diagnosis.

### 2.4.2. Selecting Relevant Regions on the Coronal Axis

Among the 35-patient data in the coronal train dataset, 35 images were selected during the selection of relevant regions. Images selected for ACL and meniscus regions detection were set at (80,200) dimensions. Figure 11 indicates the manually selected regions. The original and marked images in the selected data shifted by five pixels three times to the left and twice to right; then, a new list of six images or each image and 210 images for a total of 35 images produced was scrolled up three times by 10 pixels and down three times by 10 pixels, and a dataset of 1470 images was obtained.

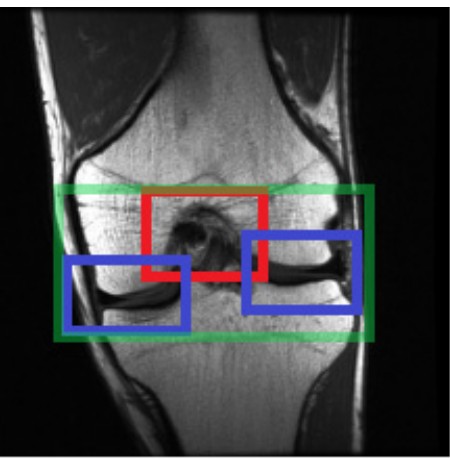

**Figure 11.** The red rectangle is the area to be selected for ACL, while the blue rectangle is the area to be selected for meniscus. The green area shows the total area that we marked for diagnosis on the coronal plane.

### 2.4.3. Selecting Relevant Regions on the Axial Axis

Among the 30-patient data in the axial train dataset, 30 images for the ACL and 30 images for the meniscus were selected during the selection of relevant regions, and these two circumstances went through a training procedure separately. The images selected for the ACL procedure were set as (100, 100) dimensions, and the images selected for the meniscus were determined in (150, 180) dimensions. Figure 12 indicates the manually selected regions. The original and marked images in the ACL data shifted four times by eight pixels to the left and right. After then, a new list of nine images or each image and 270 images for a total of 30 images produced was scrolled up once by 20 pixels and down once by 20 pixels, and a dataset of 810 images was obtained. The original and marked images in the Meniscus data shifted four times by four pixels to the left and right. Then, a new list of nine images for each image and 270 images for a total of 30 images produced was scrolled up once by 15 pixels and down once by 15 pixels, and a dataset of 810 images was obtained.

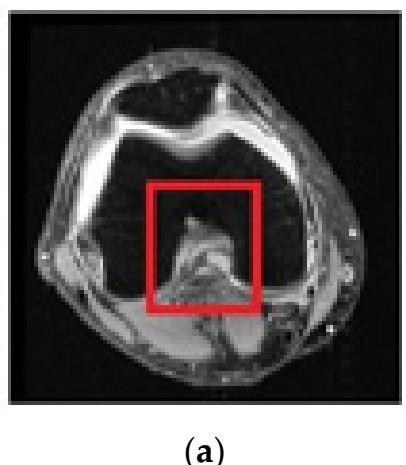    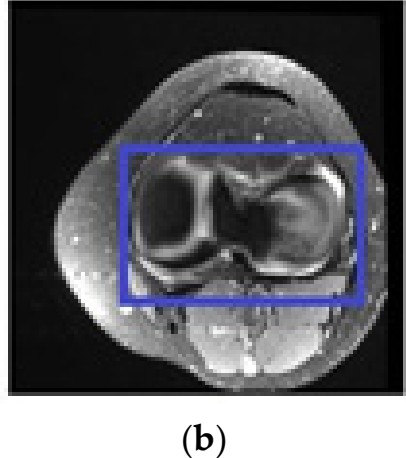

(a)                                   (b)

**Figure 12.** (**a**) The red square indicates the area selected for ACL diagnosis; (**b**) the blue square indicates the area selected for meniscus diagnosis.

### 2.4.4. Structure of the Region of Interest Model

The images selected by Model 1 among the sagittal, coronal and axial images in the designed model arrive separately and are first processed through the CNN network before being flattened and entering the denoising autoencoder. The dataset containing the selected

regions is delivered to the output by following the flattening procedure when producing the dataset in the output section. The conv2d process with 5 layers and 64 filters of (3, 3) each was used in the CNN layer. The activation function in these layers was selected as 'relu.' Maxpooling with filter dimension (2, 2) and a stride value (2, 2) was applied and padding of 'same' was set among each of these five layers. The flattening procedure on the last part of the CNN network produces a set of 10,816 elements. Following this, a denoising autoencoder with, respectively, 512, 256, 128, 256, and 512 dense layers is used. The activation function in these layers was selected as 'relu.' The images of the dimension (256 × 256 × 1) are generated by selecting regions when producing the dataset. Using the flattening procedure produces a layer of 65,536 elements. The sigmoid activation function was employed in the final layer of the denoising autoencoder, which has 65,536 elements. Figure 13 shows the model's structure as well as its parameter values.

*2.5. Diagnosis*

In Model 1, the number of images selected for the sagittal ACL ranges from 0 to 10. If the number of selected values is less than 10, they are added one after the other, starting with the first selected image, to produce an image with dimensions of (600, 100, 3).

The number of images selected for the sagittal meniscus ranges from 0 to 24. If the number of selected values is less than 24, the process is repeated, starting from the beginning, to produce an image with dimensions of (1200, 100, 3). Then, the image is converted to (600, 200, 3) dimensions. The goal here is to be compatible with ACL dimensions on the vertical axis because the images used to diagnose a sagittal abnormality are produced by merging ACL and meniscus images, and their dimensions are (600, 300, 3).

In Model 1, the number of images selected for the coronal plane ranges from zero to nine. If the number of selected values is less than nine, the process is repeated, starting from the value zero, to produce an image with dimensions of (720, 200, 3).

The number of images selected by Model 1 for ACL diagnosis in axial images ranges from zero to nine. The number of selected values that are less than nine are repeated to produce an image with dimensions of (900, 100, 3). The number of selected images in coronal images ranges from zero to six and the same repetition process is applied to produce an image with dimensions of (900, 180, 3). The images with dimensions of (900, 280, 3) are produced by merging data from ACL and meniscus for controlling abnormality.

The ACL, meniscus, and abnormal numbers of sagittal, coronal, and axial images in the MRNet dataset are disproportionate. Instead of utilising data augmentation, the weight values were manually set during the training by applying sample weight values to eliminate the disproportion. Table 7 indicates the number of positive and negative values utilised, as well as the sample weights applied prior to the dataset of the diagnostic phase.

**Table 7.** The number of positive and negative values and sample weights.

| Plane | Task | Train Positive | Train Negative | Sample Weights Positive—Negative | Validation Positive | Validation Negative |
|-------|------|----------------|----------------|----------------------------------|---------------------|---------------------|
| Sagittal | ACL | 392 | 732 | 4.46–1.0 | 51 | 67 |
| | Meniscus | 206 | 918 | 1.87–1.0 | 54 | 66 |
| | Abnormal | 907 | 217 | 1.0–4.18 | 93 | 25 |
| Coronal | ACL | 207 | 916 | 4.43–1.0 | 54 | 66 |
| | Meniscus | 395 | 728 | 1.84–1.0 | 52 | 68 |
| | Abnormal | 906 | 217 | 1.0–4.18 | 95 | 25 |
| Axial | ACL | 208 | 920 | 4.42–1.0 | 54 | 65 |
| | Meniscus | 396 | 732 | 1.85–1.0 | 52 | 67 |
| | Abnormal | 911 | 217 | 1.0–4.20 | 95 | 24 |

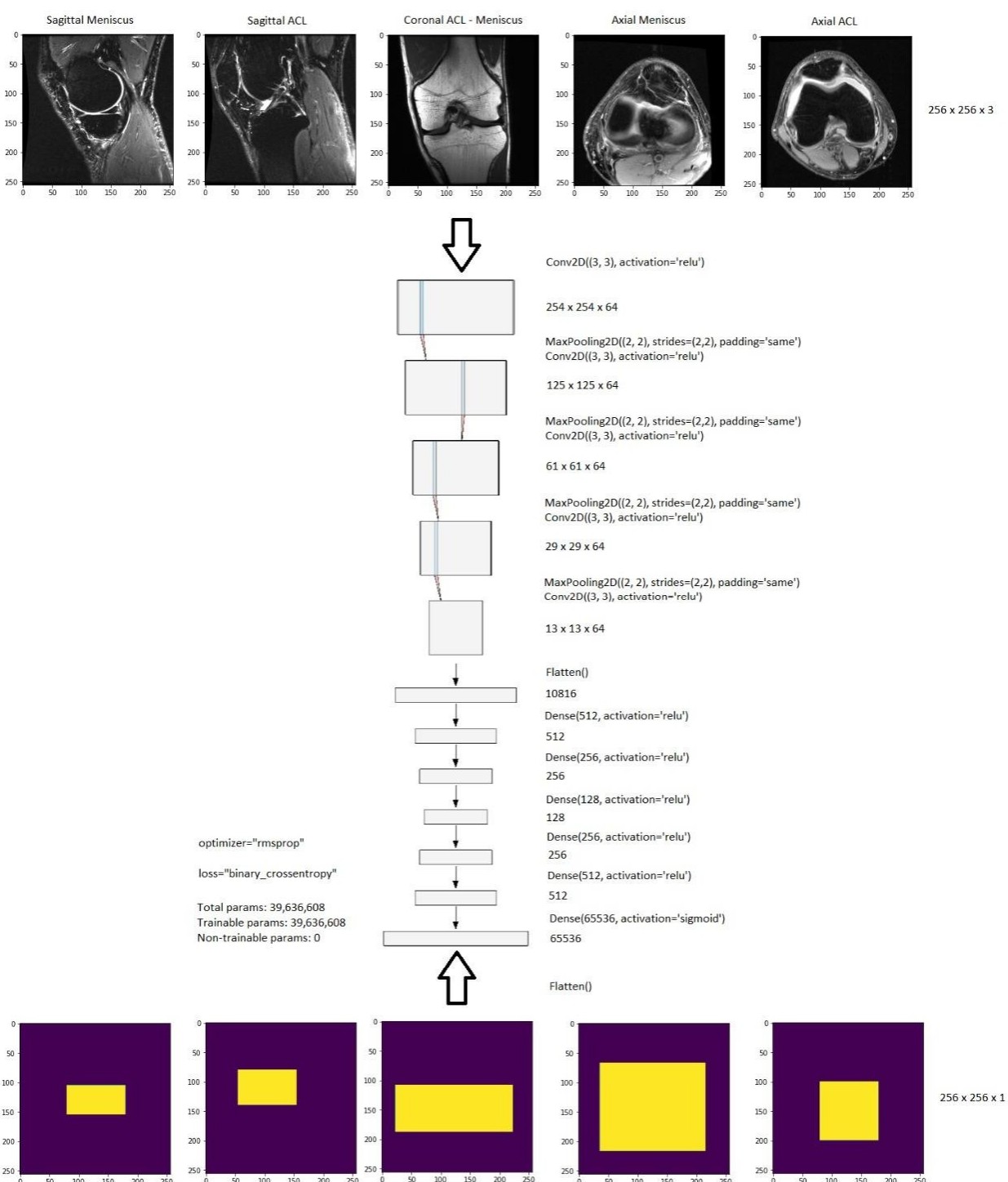

**Figure 13.** General structure of region of interest model.

ResNet50 was used during the diagnostic phase. The weight values were taken from ImageNet, and global average pooling was performed. The layer trainable was set to false. The dense layers with 512 and 128 units of relu activation function were added after the output layer. Figure 14 depicts the input sample shapes as well as the model's overall structure.

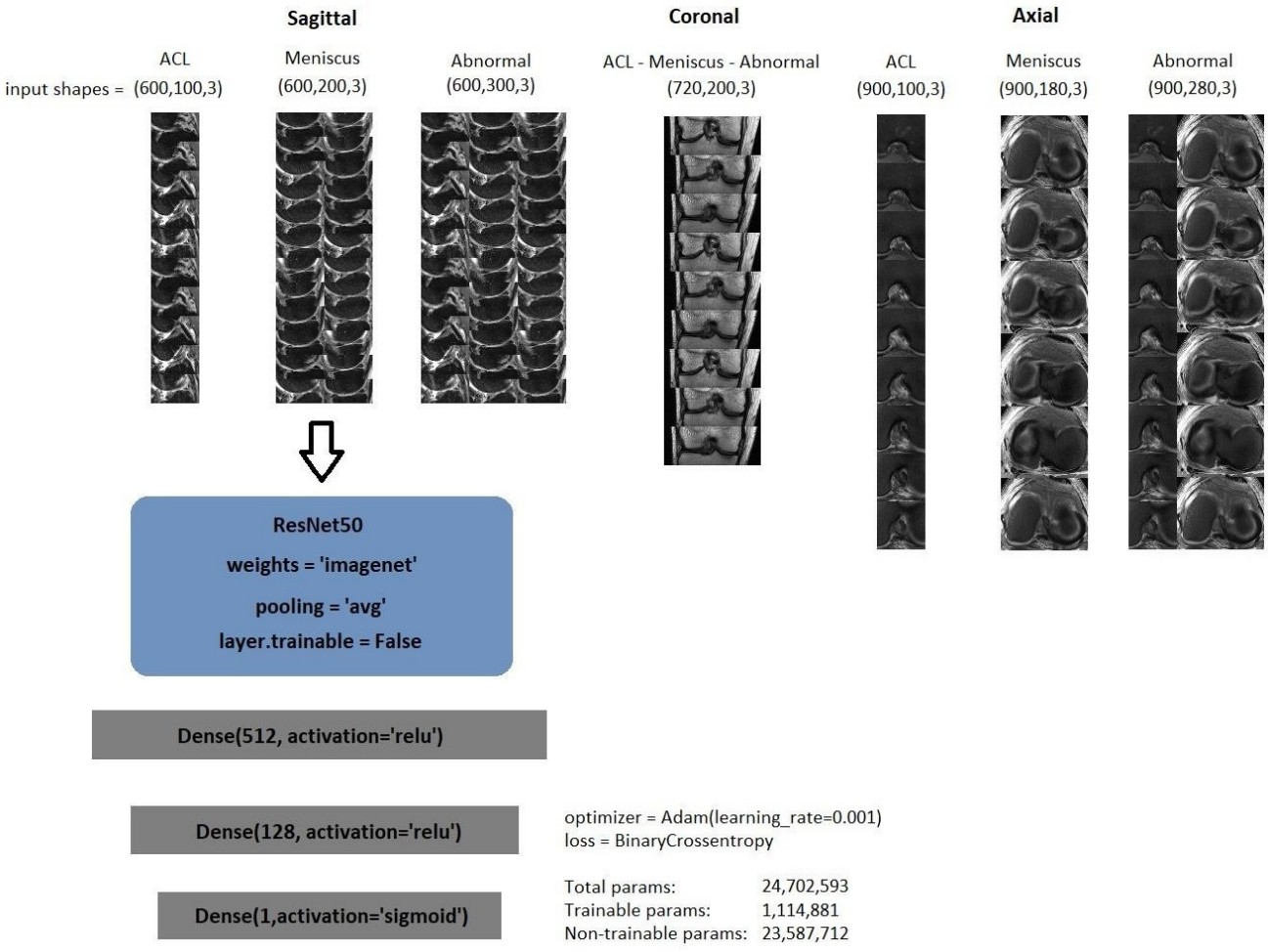

**Figure 14.** General structure of diagnosis model.

### 2.6. Structure of Progressively Operating Model

If we repeat the progressive deep learning approach for diagnosing discomfort and abnormalities in the knee area using MRI images in an overview, the MR images of each patient are captured and processed through the classification algorithm we call Model 1 in the dataset that contains sagittal, coronal, and axial images. This area is where images eligible for diagnosis are selected, and the selected images are fed into a model called Model 2 that combines the deep learning approaches of CNN with a denoising autoencoder to identify the regions where the disease will be diagnosed. After the regions are selected, they are scaled and sent to the ResNet50 model, which we refer to as Model 3, for diagnosis. Model 3 employs more dense layers than Model 1, uses a different optimiser, and the layer in the ResNet50 model is set to false. This is how the differences between the ResNet50 models, employed in Model 1 and Model 3, can be distinguished. Figure 15 depicts the schematic of the progressive deep learning approach.

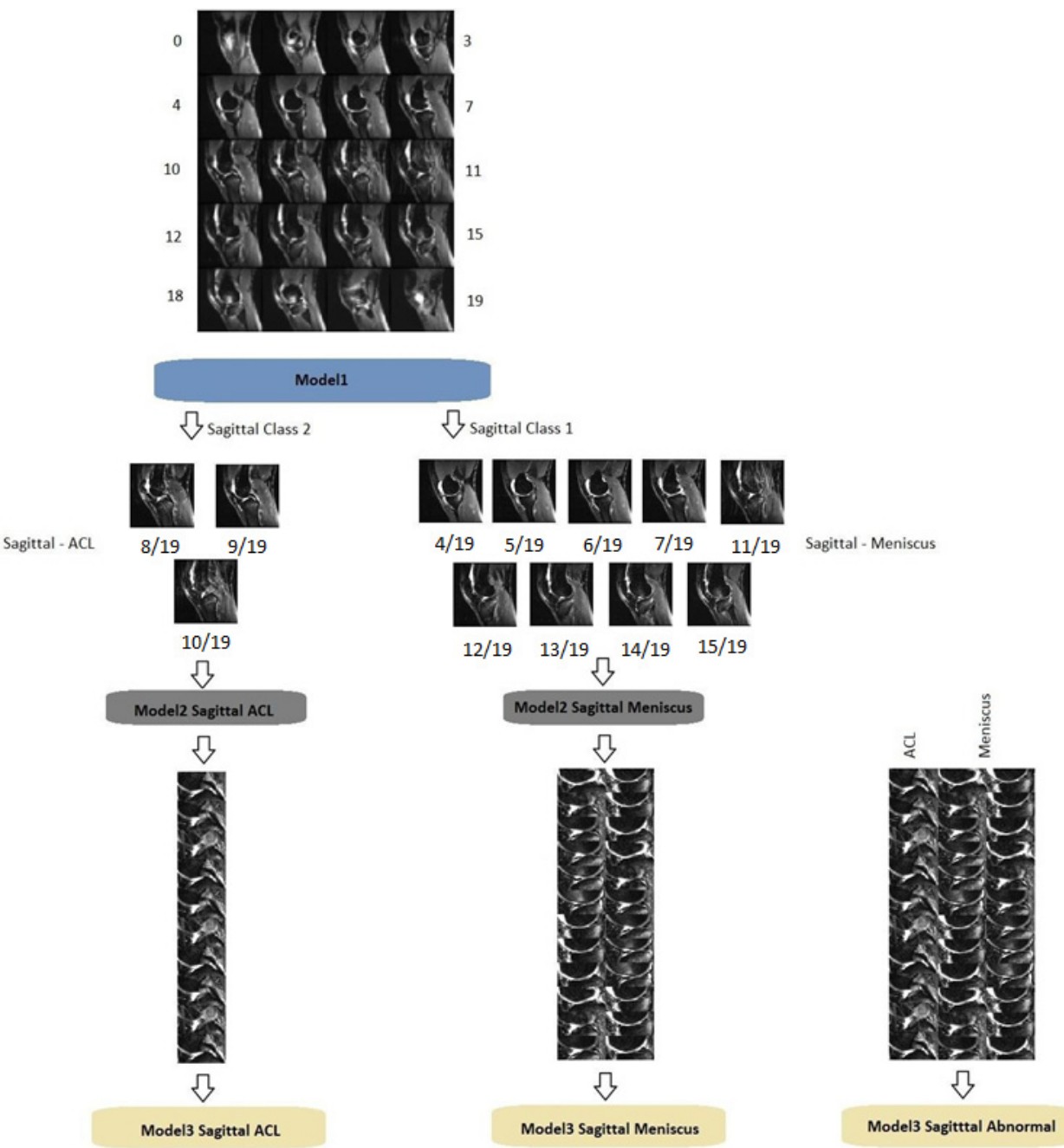

**Figure 15.** General structure of progressively operating model.

## 3. Results

With our progressive deep learning model, we classified the MRI to identify whether there is an ACL, meniscus tear and abnormal disorders in the sagittal, coronal and axial axes, selected the relevant images, located the region to be diagnosed and diagnosed the disease within the selected images. To evaluate the performance of the studies, we found the accuracy, sensitivity, specificity, Matthew's correlation coefficient (MCC) and the area under the receiver operating characteristic curve (ROC-AUC) values.

Accuracy is the ratio between the number of correctly classified samples and the total number of samples. The following four circumstances can develop when someone is classified as healthy or sick for a disease:

- Classification of sick people as sick—"True positive".
- Classification of healthy people as sick—"False positive".

- Classification of healthy people as healthy—"True negative".
- Classification of sick people as healthy—"False negative".

Sensitivity is the ratio of the number of true negatives to the sum of the true positive and false negative numbers. Specificity is the ratio of the number of true negatives to the sum of true negative and false positive numbers [53]. An MCC is an effective method when the number of dataset elements is disproportionate—that is, the number of instances in one class is much greater than the number of instances in other classes [54]. The ROC is a probability curve, and the AUC represents the degree of differentiability of facts and errors as an area below this curve [55].

We discovered that no images were selected from certain patients during the first part of the study while selecting acceptable images for diagnosis. Therefore, the data for these patients were not included in the training and validation dataset. Table 8 provides the number of data used in the train and validation dataset and the results we achieved during the evaluation.

**Table 8.** The number of data used in the train and validation dataset and the results.

| Plane | Train | Validation | Task | Accuracy | Sensitivity | Specificity | MCC | ROC-AUC |
|-------|-------|-----------|------|----------|-------------|-------------|-----|---------|
| | | | ACL | 0.7881 | 0.5741 | 0.9688 | 0.6025 | 0.8947 |
| Sagittal | 1124 | 118 | Meniscus | 0.7712 | 0.5686 | 0.9254 | 0.5403 | 0.7987 |
| | | | Abnormal | 0.8898 | 0.9462 | 0.68 | 0.6571 | 0.9316 |
| | | | ACL | 0.7583 | 0.8519 | 0.6818 | 0.5346 | 0.8297 |
| Coronal | 1123 | 120 | Meniscus | 0.75 | 0.7115 | 0.7794 | 0.4910 | 0.7393 |
| | | | Abnormal | 0.8667 | 0.9473 | 0.56 | 0.5644 | 0.8029 |
| | | | ACL | 0.8319 | 0.7222 | 0.9231 | 0.6655 | 0.8721 |
| Axial | 1128 | 119 | Meniscus | 0.6891 | 0.7115 | 0.6716 | 0.3801 | 0.7075 |
| | | | Abnormal | 0.8992 | 0.9579 | 0.6667 | 0.6702 | 0.8596 |

## 4. Discussion

In this study, we examined how to configure a deep learning architecture to locate faulty data in a dataset of MR images of arrays, select the region for diagnostics, and enhance classification performance. From MR images, we designed sequential models for detecting ACL tears, meniscus tears, and abnormalities in the knee.

The study by Bien et al. [17] found the ROC-AUC value of 0.826 in the detection of a meniscus tear, 0.956 in the detection of an ACL tear, and 0.936 in the detection of abnormality. The researchers employed the logistic regression-based ensemble learning approach to build a common disturbance detection model in the coronal, sagittal, and axial axes. Our study excluded noisy and/or damaged images from the dataset at the Model 1 stage since they were unsuitable for diagnosis. Therefore, the joint decision model could not be employed in our study, as the number of elements in the dataset differs for each axis and each disturbance detection. The study by Tsai et al. [33] utilised coronal axis images to diagnose meniscus tears and axial axis images to diagnose ACL tears, as well as abnormalities. The ROC-AUC values acquired from the study were 0.904 for meniscus, 0.96 for ACL and 0.941 for abnormality diagnosis. They achieved higher results in three of the nine different stages than the findings of our study. Our study, by employing the efficiently-layered network (ELNet) rather than the ResNet50 model in the Model 3 section, can be re-evaluated. The study by Azcona et al. [18] employed a logistic regression-based ensemble learning approach. The ROC-AUC values acquired from the study were 0.9557 for ACL, 0.9081 for meniscus, and 0.9381 for abnormality diagnosis. Evaluating our study generally, we classified by viewing the images given in the articles reviewed at the ACL section in the Model 1 stage and the related studies section at the meniscus classification stage. It is believed that the success rate would improve if radiologists supported the classification stage.

## 5. Conclusions

In this study, we studied three different planes: coronal, sagittal and axial. The study differs from other studies in the literature as follows.

- For the first time, regions were classified and selected among the MR images in the diagnosis of knee problems, and successful results were achieved at the classification stage.
- The region of interest study was carried out in previous studies. However, convolutional neural networks and denoising autoencoders were employed for the first time to carry out a diagnostic study and were successful in detecting the region.
- Since our study goes through several deep learning models sequentially, it would provide later findings than other studies.
- Other study techniques, when undertaken in a hospital setting, may produce erroneous results since it is impossible to detect whether the image is damaged or over noisy.

As a result, our work was accomplished by employing a progressive deep learning method that can be utilised to detect erroneous or over noisy radiological data. In addition, it is believed that our work could be instrumental to boost the efficiency of our radiological diagnoses on the detection of the region and the diagnosis of disease with artificial intelligence. Following this study, the goal should be to divide the diseases detected by MR into diagnostic sections, focusing on the region where the diagnosis would be made by employing the model we designed.

The location of the tumour region is critical to diagnose cancer from MRI image indexes. Breast cancer is classified by the regions where tumours are discovered, and the incidence of cancers varies depending on the part of the breast [56,57]. There are also several studies based on the locations of tumours in the brain [27,58]. When diagnosing benign and malignant tumours from MRI image indexes, the image range to be diagnosed in the diagnostic image index can be selected and the suitability for diagnosis can be analysed according to the noise condition if the regions with tumours in the dataset are marked and trained with Model 1. Marking the location of the tumour on a single image and training Model 2, which we built, can contribute to the determination of the location. Model 3 can be employed to diagnose malignant or benign tumours following these sequential procedures.

**Author Contributions:** Conceptualisation, A.C.K. and F.H.; methodology, A.C.K. and F.H.; software, A.C.K. and F.H.; validation, A.C.K. and F.H.; formal analysis, A.C.K. and F.H.; investigation, A.C.K. and F.H.; resources, A.C.K. and F.H.; data curation, A.C.K. and F.H.; writing—original draft preparation, A.C.K. and F.H.; writing—review and editing, A.C.K. and F.H.; visualisation, A.C.K. and F.H.; supervision, F.H.; project administration, A.C.K. and F.H. All authors have read and agreed to the published version of the manuscript.

**Funding:** This research received no external funding.

**Institutional Review Board Statement:** Not applicable.

**Informed Consent Statement:** Not applicable.

**Data Availability Statement:** Data used in this study are available at https://stanfordmlgroup.github.io/competitions/mrnet (accessed on 9 November 2021).

**Conflicts of Interest:** The authors declare no conflict of interest.

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
