# Peer review of "Detection and Classification of Knee Injuries from MR Images Using the MRNet Dataset with Progressively Operating Deep Learning Methods"

_make, doi:10.3390/make3040050_

Round 1

Reviewer 1 Report

This submission presents the whole procedure of detecting knee injuries from MR images, from data selection to diagnosis. In each step an effective network is adopt to achieve the corresponding goal. The whole study is interesting, but the paper should be significantly improved before being considered for publication.

A major issue is the writing of the paper. Extensive editing of English language and style is recommended. Expressions like "provided making the diagnosis" in line 19 are confusing. Acronyms should be expanded or explained at the first appearances, even if they are widely used in the field.

Since the whole procedure is basically combining existing techniques and the novelty of the paper is somewhat limited, it gets more important to review and discuss recent related works that use deep learning for disease detection, e.g., "Adaptive Squeeze-and-Shrink Image Denoising for Improving Deep Detection of Cerebral Microbleeds" in MICCAI'21.

Besides, in line 27, it might be controversial to say "for a long time deep learning approaches have been adopted ...", since when we say "a long time" we usually mean decades, while in fact the first use of deep learning in MRI diagnosis is just a few years ago.

Author Response

Your comments have been very helpful to me. Thank you so much. Please see the attachment.

Kind regards,
Ali Can Kara

Reviewer 2 Report

The authors did an interesting report with possible future applications. However, few minor revisions are needed:

In the section 2.2. Related work, discussion of related work in NNs should be expanded with more recent work. More references to diagnosis papers with NNs should be included, like ”Towards Accurate Diagnosis of Skin Lesions Using Feedforward Back Propagation Neural Networks, Diagnostics 2021, 11, 936, https://doi.org/10.3390/diagnostics11060936” and ”Lesion Classification Based on Surface Fractal Dimensions and Statistical Color Cluster Features Using an Ensemble of Machine Learning Techniques. Cancers 2021, 13(21), 5256. https://doi.org/10.3390/cancers13215256”

The motivation of the approach with NNs needs further clarification

line 125- please explain this duplication process

line 128- The abbreviations should be explained at their first in text!

It is well known that SGD performs frequent updates with a high variance that cause the objective function to fluctuate heavily. Please explain how this this shortcoming is overcome

Table 1 Classes 0 and 1 should be defined

Tables 3 and 5 In a similar way, the meaning of these classes should be presented

Could this method save time or could it be defined as a method to be used only for specific knee injuries?

A section devoted to the Conclusion should be added. In this section, please add sentences about advantages or disadvantages of the proposed method

Author Response

(The authors gave the same response as above.)

Reviewer 3 Report

The work is welll written and the paper is well prepared - good tables, figures, illustrating matter. However, and despite the good state-of-art discussion, the work does not introduce a comparison between the authors' results and the known approaches. A table should be given to show how the authors' methods outperform thje others' algorithms and/or methodologies.   

Author Response

(The authors gave the same response as above.)

Round 2

Reviewer 1 Report

The authors have adequately addressed the reviewers' concerns and the quality of the paper has been improved, so I recommend to accept the paper for publication in MAKE.